# Q-SFT: Q-Learning for Language Models via Supervised Fine-Tuning

**Joey Hong**          **Anca Dragan**          **Sergey Levine**
University of California, Berkeley
{jxihong,anca,svlevine}@berkeley.edu

## Abstract

Value-based reinforcement learning (RL) can in principle learn effective policies for a wide range of multi-turn problems, from games to dialogue to robotic control, including via offline RL from static previously collected datasets. However, despite the widespread use of policy gradient methods to train large language models for single turn tasks (e.g., question answering), value-based methods for multi-turn RL in an off-policy or offline setting have proven particularly challenging to scale to the setting of large language models. This setting requires effectively leveraging pretraining, scaling to large architectures with billions of parameters, and training on large datasets, all of which represent major challenges for current value-based RL methods. In this work, we propose a novel offline RL algorithm that addresses these drawbacks, casting Q-learning as a modified supervised fine-tuning (SFT) problem where the probabilities of tokens directly translate to Q-values. In this way we obtain an algorithm that smoothly transitions from maximizing the likelihood of the data during pretraining to learning a near-optimal Q-function during finetuning. Our algorithm has strong theoretical foundations, enjoying performance bounds similar to state-of-the-art Q-learning methods, while in practice utilizing an objective that closely resembles SFT. Because of this, our approach can enjoy the full benefits of the pretraining of language models, without the need to reinitialize any weights before RL finetuning, and without the need to initialize new heads for predicting values or advantages. Empirically, we evaluate our method on both pretrained LLMs and VLMs, on a variety of tasks including both natural language dialogue and robotic manipulation and navigation from images.

## 1 Introduction

Recently, some of the most impressive feats in AI have been performed through language models, which are pretrained on large-scale data and adapted to a wide range of downstream tasks (Bommasani et al., 2021). Many of these tasks, such as natural language dialogue or robotic control, require complex sequential decision-making. Reinforcement learning (RL) Sutton & Barto (2018) is a powerful paradigm for solving such tasks (Mnih et al., 2013; Silver et al., 2017; AlphaStar, 2019). Furthermore, offline RL Levine et al. (2020) has been shown to do so from only static datasets, such as suboptimal demonstrations from any unknown behavior policy, without the need for any additional interaction. Though offline RL has been used to fine-tune large language models (LLMs) or vision language models (VLMs) (Ouyang et al., 2022; Bai et al., 2022b), its usefulness has been limited to generating better single responses rather than multi-turn, sequential scenarios where RL should theoretically shine. For example, across various dialogue tasks, offline RL fine-tuning of LLMs does not reliably outperform supervised fine-tuning (SFT) (Sodhi et al., 2023; Abdulhai et al., 2023). Furthermore, in the realm of navigation and control, popular VLMs are still fine-tuned for multi-task control using SFT (Brohan et al., 2023b;a; Collaboration et al., 2024).

Single-turn problems, such as answering questions, can be tackled with policy gradient methods (Ouyang et al., 2022; Rafailov et al., 2023), but sequential or *multi-turn* problems, such as dialogue or robotic control, require sample-efficient methods that can utilize data to reason about the dynamics of the problem, which typically requires training value functions (Abdulhai et al., 2023; Hong et al., 2023). This is in multi-turn problems, the agent must plan their actions to optimize some long-term objective. Although there are many effective value-based RL methods that could be applied to

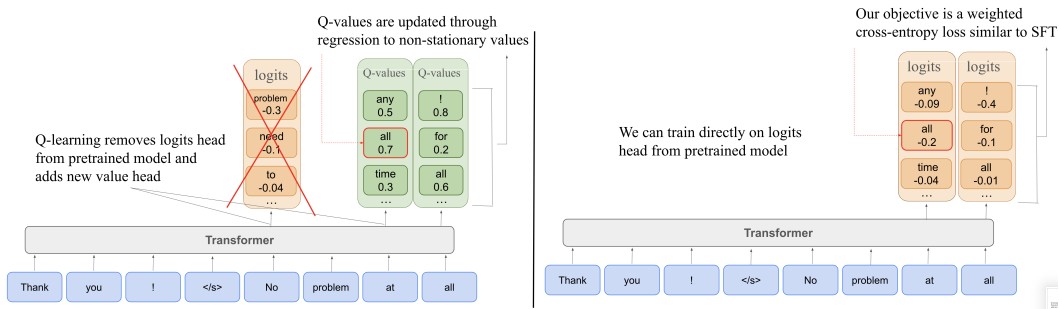

Figure 1: Our proposed approach allows us to directly leverage the logits from a pretrained model to train value functions. Prior approaches require separately initializing a value head.

LLMs and VLMs, in practice such methods have been difficult to adapt to these models with the same effectiveness as policy gradients. We posit that this is due in part to a mismatch between the pretraining objective that these models use, i.e. maximum likelihood estimation, and the fine-tuning objective necessary to train value functions. This discrepancy means that fine-tuning using multi-turn RL may require discarding some of knowledge gained by maximum likelihood pretraining of LLMs and VLMs, including a broad understanding of language, vision, and even sequential reasoning.

Specifically, we hypothesize two reasons for why fine-tuning foundation models using offline RL is unsuitable in practice. First, typical offline RL methods require regressing value functions that estimate how appropriate actions, such as an utterance in dialogue, are. Such algorithms, known as Q-learning, have achieved impressive results when applied on small networks (AlphaStar, 2019; Mnih et al., 2013), but surprisingly attain disappointing performance when scaled to larger ones (Sodhi et al., 2023). Recent work has attributed this lack of scaling to instability in the value-learning objective, namely in regression towards non-stationary values (Farebrother et al., 2024). More importantly, a major advantage of SFT is the potential to leverage existing capabilities of large pretrained models to drastically improve the efficiency when learning a new downstream task. However, language models are trained to predict likelihoods, but Q-learning instead aims to predict action values; therefore, when fine-tuning, Q-learning algorithms discard the learned likelihoods in favor of only utilizing the underlying representations, which eliminates some of the useful prior knowledge within the pretrained models. We illustrate this in Figure 1, where value functions are trained must be trained via a new head with reset weights.

In this work, we propose a new algorithm that remedies both drawbacks. Our key insight is simple: *by adding weights to the traditional supervised fine-tuning objective, we can learn probabilities that conservatively estimate the value function instead of the behavior policy*. In practice, our approach is implemented by adding weights to the maximum likelihood objective, yielding a *weighted cross entropy loss* where weights are target action values computed from the Bellman recurrence relations. By using this objective, we are able to avoid the unstable regression objective commonly used in value learning, as well as directly leverage the initial likelihoods resulting from large-scale pretraining. Theoretically, we can show that such objective results in learned likelihoods that are a product of the data distribution and Q-values, and that our approach is principled and results in performance bounds competitive with other state-of-the-art approaches. Empirically, we demonstrate the effectiveness of our method on a variety of tasks involving both LLMs, such as language games and dialogue, as well as VLMs, such as navigation and robotic manipulation.

## 2    RELATED WORK

Much of the recent work on reinforcement learning (RL) finetuning of LLMs and VLMs uses policy gradient methods and reward models learned from human feedback (e.g., RLHF) (Ziegler et al., 2020; Stiennon et al., 2020; Wu et al., 2021; Nakano et al., 2022; Bai et al., 2022a; Christiano et al., 2023; Rafailov et al., 2023), or from handcrafted AI systems (e.g., RLAIF) (Bai et al., 2022b), to generate better responses to various queries. However, there is a large discrepancy in the capabilities required to perform self-contained responses in single-step tasks, such as question-answering, and responses in a multi-turn scenarios, such as dialogue. Namely, the latter requires planning to optimize a long-term

objective,. Various prior works provide evidence that existing fine-tuning methods are insufficient to enable language models with such planning capabilities (Bachmann & Nagarajan, 2024).

In principle, value-based RL (Lange et al., 2012; Levine et al., 2020), specifically Q-learning, can learn effective policies for multi-step tasks that outperform pure imitation via supervised fine-tuning (Kumar et al., 2022). Many offline RL algorithms exist that reap the benefits of value-based RL using only static datasets, such as those currently used to fine-tune language models. Though offline RL algorithms require handling distribution shift (Kumar et al., 2019), where the learned policy selects out-of-distribution (OOD) actions with unpredictable consequences, many methods exist that effectively tackle this challenge (Kumar et al., 2020; Kostrikov et al., 2021; Kidambi et al., 2020; Yu et al., 2020; 2021). Due to the promising benefits of offline RL on learning from demonstrations, algorithms have been proposed for learning LLM policies to some success in robotic manipulation (Chebotar et al., 2023) and language tasks (Snell et al., 2022). However, recent evaluation has shown that, on a variety of natural language tasks, Q-learning approaches are often outperformed by supervised ones (Sodhi et al., 2023; Abdulhai et al., 2023). We hypothesize this is due to the mismatch between value-based RL fine-tuning and maximum likelihood pretraining, and propose a new approach that remedies this core issue.

There also exist a paradigm of supervised approaches called return conditioned supervised learning (RCSL), which learn conditional policies on return via a supervised learning objective (Brandfonbrener et al., 2022). The most notable algorithm is Decision Transformer (DT) (Chen et al., 2021), which can train LLM policies that outperform traditional offline RL methods that rely on Q-learning. Though it performs well in practice, there is theoretical evidence that the ceiling of performance of such algorithms is below that of value-based offline RL. Specifically, Brandfonbrener et al. (2022) showed that DT and similar approaches can only identify the optimal policy under stronger conditions on the offline data than value-based RL. Our proposed algorithm is similar to RCSL in that we also use a maximum likelihood loss, but we learn values and reap the theoretical benefits of other value-based methods.

Recently, prior attempts have also been made to improve value-based RL algorithms for fine-tuning language models. Chebotar et al. (2023) propose Q-learning with transformer value functions in manipulation and control tasks by converting actions to sequences of tokens. We adopt their insight when evaluating on robotics tasks, but use a fundamentally different objective to learn values. Most similar to ours, Farebrother et al. (2024) propose to replace the regression loss from Q-learning with a cross-entropy loss by casting value learning as a classification problem. However, while the proposed method also converts value functions to distributions, these likelihoods are not naturally derived from the logits obtained from large-scale pretraining, and must instead be learned from scratch via a separate head with reset weights. Therefore, like traditional Q-learning, they also suffer from being unable to leverage pretraining efficiently, unlike our approach whose likelihoods are directly initialized by the logits of pretrained LLMs or VLMs.

## 3 PRELIMINARIES

Our work proposes a new RL algorithm for fine-tuning language models, specifically for multi-turn tasks such as dialogue or manipulation and control. Language models operate over a discrete vocabulary of tokens $\mathcal{V}$, and are trained to maximize the likelihood the best next-token $x_{m+1}$ given an input sequence $(x_0, \ldots, x_m)$ of tokens, given by $\pi(x_{m+1}|x_0, \ldots, x_m)$. In a multi-turn task such a dialogue, the tokens are words that are chained to form utterances, and the best next-token requires complex, sequential reasoning to understand the utterances so far and plan for the next one. Traditionally, this kind of reasoning can be learned via reinforcement learning (RL).

**RL fundamentals.** RL aims to optimize agents that interact with a Markov Decision Process (MDP) defined by a tuple $(\mathcal{S}, \mathcal{A}, P, r, \mu_1, \gamma)$, where $\mathcal{S}$ represents the set of all possible states, $\mathcal{A}$ is the set of possible actions, $\mu_1$ is the initial state distribution, and $\gamma$ is the discount factor. When action $a \in \mathcal{A}$ is executed at state $s \in \mathcal{S}$, the next state is generated according to $s' \sim P(\cdot|s, a)$, and the agent receives stochastic reward with mean $r(s, a) \in [0, 1]$.

The Q-function $Q^\pi(s, a)$ for a policy $\pi(\cdot|s)$ represents the discounted long-term reward attained by executing $a$ given observation history $s$ and then following policy $\pi$ thereafter. $Q^\pi$ satisfies the

Bellman recurrence:

$$Q^\pi(s,a) = r(s,a) + \gamma \mathbb{E}_{s' \sim P(\cdot|s,a), a' \sim \pi(\cdot|s')} \left[ Q^\pi(s',a') \right] .$$

The value function $V^\pi$ considers the expectation of the Q-function over the policy $V^\pi(h) = \mathbb{E}_{a \sim \pi(\cdot|s)} \left[ Q^\pi(s,a) \right]$. Meanwhile, the Q-function of the optimal policy $Q^*$ satisfies:

$$Q^*(s,a) = r(s,a) + \gamma \mathbb{E}_{s' \sim P(\cdot|s,a)} \left[ \max_{a'} Q^*(s',a') \right] ,$$

and the optimal value function is $V^*(s) = \max_a Q^*(s,a)$. Finally, the expected cumulative reward is given by $J(\pi) = \mathbb{E}_{s_1 \sim \mu_1} \left[ V^\pi(s_1) \right]$. The goal of RL is to optimize a policy $\pi(\cdot \mid s)$ that maximizes the cumulative reward $J(\pi) = \mathbb{E}_{\mu_1} \left[ V^\pi(s_1) \right]$.

In offline RL, we are provided with a dataset $\mathcal{D} = \{(s_i, a_i, r_i, s'_i)\}_{i=1}^N$ of size $|\mathcal{D}| = N$. We assume that the dataset $\mathcal{D}$ is generated i.i.d. from an effective behavior policy $\pi_\beta(a|s)$. Many state-of-the-art offline RL methods build on Q-learning, which trains a Q-function using parameters $\theta$ on dataset $\mathcal{D}$ by minimizing temporal difference (TD) error.

$$\mathcal{L}_{TD}(\theta) = \mathbb{E}_{(s,a,r,s') \sim \mathcal{D}} \left[ \left( r + \gamma \max_{a'} Q_{\bar\theta}(s',a') - Q_\theta(s,a) \right)^2 \right] , \tag{1}$$

where $Q_\theta(s,a)$ is the parameterized Q-function, and $\bar\theta$ parameterize a target network and is a slow-moving copy of $\theta$.

**RL for language generation.** Language generation can be viewed as an MDP, where states are sequences of tokens from a finite vocabulary $\mathcal{V}$ (Ramamurthy et al., 2023). All tokens that the agent initially observes are used as our initial state, $s_0 = (x_0, \ldots, x_m)$, where $x_i \in \mathcal{V}, \forall i \in [m]$. At timestep $t$, an action $a_t \in \mathcal{V}$ is some token in the vocabulary. As long as $a_t$ is not a special end-of-sequence <EOS> token, the transition function deterministically appends $a_t$ to state $s_t$ to form $s_{t+1}$. Otherwise, the agent observes (potentially stochastic) responses from the environment, i.e. utterances by conversational partners in the case of multi-turn dialogue, $o_t = (y_0, \ldots, y_n)$, which also consist of tokens in the vocabulary; then, the transition function appends both $a_t$ and responses $o_t$ to state $s_t$. This continues until the last timestep $T$ where we obtain a state $s_T$ and the agent receives a deterministic reward $r(s_T)$.

It becomes clear that a policy $\pi(a|s)$ is a language model that parses all the language tokens seen so far as the state, and computes a distribution over tokens as the next action to take. Recently, RL has been considered for learning policies that are LLMs or VLMs for difficult tasks such as generalist robotic manipulation or dialogue. Because value learning is very different from traditional next-token prediction, preforming such fine-tuning requires reparameterizing the pretrained language model, such as by adding value heads with independently initialized weights (Snell et al., 2023).

## 4 Q-LEARNING VIA SUPERVISED FINE-TUNING

We will now describe our proposed offline RL algorithm, which we dub Q-learning via Supervised Fine-tuning (Q-SFT). Concretely, instead of training value functions by fitting Q-values to their Bellman backup target via a regression loss, we instead fine-tune directly on the probabilities learned from large-scale pretraining —like in SFT— via a *weighted cross-entropy* loss, such that the resulting probabilities also capture the desired Q-values.

### 4.1 LEARNING VALUES AS PROBABILITIES

Recently, large neural networks such as LLMs and VLMs have been successfully trained and fine-tuned on demonstration data using supervised learning. If we adopt the earlier multi-turn formalism in Section 3 and view these models as agents, such approaches train a policy $\pi_\phi(a|s)$ with parameters $\phi$ by minimizing cross-entropy loss:

$$\mathcal{L}_{CE}(\phi) = \mathbb{E}_{(s,a) \sim \mathcal{D}} \left[ \log \pi_\phi(a \mid s) \right] . \tag{2}$$

Because the resulting policy approximates the behavior policy $\pi_\phi(a|s) \approx \pi_\beta(a|s)$, this approach has also been dubbed behavioral cloning (BC). While BC scales well to complex tasks and networks, the

resulting policy can only be as good as the behavior policy, which is insufficient when the dataset is not curated from expert demonstrations.

In contrast, Q-learning enables the learned policy to greatly outperform the behavior policy (Kumar et al., 2022), by instead having the policy behave according to the estimated Q-values. This can be done via *policy extraction*, such as $\pi(a|s) = \mathbb{1}[a = \arg\max_a' Q_\theta(s, a')]$ or the entropy-regularized variant $\pi(a|s) \propto \exp(Q_\theta(s, a))$. However, as alluded to earlier, the Q-function $Q_\theta(s, a)$ cannot be naturally derived from pretrained language models, which output probabilities, and require modifying their architectures as in Figure 1.

Our goal is to provide a way to learn Q-values for multi-turn RL problems with language models such that the Q-function can be initialized from a model pretrained via supervised learning (i.e., maximum likelihood estimation), *without* the need to reinitialize weights or add new heads to represent the Q-values. An autoregressive sequence model (e.g., a transformer) outputs the probability of each token conditioned on the past history. In order to avoid adding new heads or reinitializing weights, the Q-values have to also be represented by these same probabilities. Furthermore, to maximize transfer from pretraining, we would like our proposed *loss function* to also closely resemble the maximum likelihood loss function used for pretraining.

We propose a simple modification to the BC objective in Equation 2. Our modification hinges on the following observation. Let $p_\theta(a|s)$ represent the probability of action $a$ under state $s$, and are optimized via the *weighted* cross entropy loss

$$\mathcal{L}_{\mathrm{WCE}}(\theta) = \mathbb{E}_{(s,a)\sim\mathcal{D}} \left[ w(s, a) \log p_\theta(a \mid s) + (1 - w(s, a)) \log p_\theta(a_d \mid s) \right],$$

where $w(s, a)$ are weights, and $a_d$ is some dummy action. The resulting probabilities that optimize this objective approximate $\widehat{p}_\theta(a|s) \approx w(s, a)\pi_\beta(a|s)$ for all $a \neq a_d$. Our goal is, via a proper choice of weights, to learn probabilities that are conservative estimates of the true Q-values $\widehat{p}_\theta(s, a) \approx Q^*(s, a)$. In order to do so, we require the following assumption on bounded total rewards:

**Assumption 4.1.** *For any policy $\pi$, we have $\sum_{t=1}^\infty \gamma^{t-1} r_t \leq 1$.*

This assumption has been made by multiple prior works without loss of generality (Ren et al., 2021; Kumar et al., 2022), as rewards can, in theory, be scaled without affecting the optimal policy in the MDP. Furthermore, many tasks of interest, such as dialogue, have sparse rewards, where we observe success or failure only after the conversation has ended.

Following the above observation, let us define the *empirical Bellman probability operator* $\widehat{\mathcal{B}}^*$ for transition $(s, a, r, s')$ as

$$\widehat{\mathcal{B}}^* p_\theta(a \mid s) = r + \gamma \max_{a'} \frac{p_\theta(a' \mid s')}{\pi_\beta(a' \mid s')}.$$

Note that this is different from the traditional Bellman operator in that we additionally divide by $\pi_\beta$ in the backup. Then, we consider the following weighted cross-entropy loss:

$$\mathcal{L}_{\mathrm{WCE}}(\theta) = \mathbb{E}_{(s,a,r,s')\sim\mathcal{D}} \left[ \widehat{\mathcal{B}}^* p_{\bar{\theta}}(a \mid s) \log p_\theta(a \mid s) + \sum_{a'\neq a} \frac{1 - \widehat{\mathcal{B}}^* p_{\bar{\theta}}(a \mid s)}{|\mathcal{A}| - 1} \log p_\theta(a' \mid s) \right]. \quad (3)$$

Here, we see that our loss is an instance of weighted cross entropy loss with weights approximately equal to Bellman target values $\widehat{\mathcal{B}}^* p_{\bar{\theta}}(a|s)$. The primary difference is that instead of introducing a dummy action, we equally distribute the leftover weight across the remaining actions. As we will show, this acts as a label-smoothing term that ultimately regularizes the probabilities. We will show later that in the absence of sampling error, our learned likelihood function $\widehat{p}_\theta(a|s)$ satisfies $Q^*(s, a) \geq \widehat{p}_\theta(a|s) \geq \pi_\beta(a|s) Q^*(s, a)$. This means that we are able to effectively learn a conservative estimation of the Q-function as a likelihood, without the need for optimizing a potentially unstable and poorly-scaling TD objective.

In addition, because probabilities are modeled directly by existing language models, we do not need to modify the parameterization of such models in order to perform such fine-tuning, i.e. by resetting weights or adding a new head. Namely, our likelihood function $\widehat{p}_\theta(a|s)$ can be directly initialized from the logits of a pretrained LLM or VLM.

## 4.2 Theoretical Analysis

In the previous section, we motivated a new objective $\mathcal{L}_{\text{WCE}}$ given by Equation 3 that learns modified probabilities over actions $p_\theta$ directly from the logits of a base LLM or VLM. Here, we will show that such probabilities serve as a conservative approximation of the true Q-values. To simplify exposition, we consider a simple modification of Equation 3, where instead of empirical operator $\widehat{\mathcal{B}}^*$, we use the true operator:

$$\mathcal{B}^* p_\theta(a \mid s) = r(s,a) + \gamma \mathbb{E}_{s' \sim P(\cdot \mid s,a)} \left[ \max_{a'} \frac{p_\theta(a' \mid s')}{\pi_\beta(a' \mid s')} \right] .$$

Note that it is simple to adapt our analysis to the empirical operator. Namely, it requires obtaining high-probability bounds of the form $\forall (s,a)$, $\left| \widehat{\mathcal{B}}^* p_\theta(a \mid s) - \mathcal{B}^* p_\theta(a \mid s) \right| \leq \frac{C}{\sqrt{n(s,a)}}$ where $C$ is a constant independent of $\gamma$. This kind of inequality commonly arises in analysis of offline RL algorithms (Kumar et al., 2020; 2022).

Our main theoretical result is that our learned probabilities satisfy being conservative estimates of the true value function:

**Theorem 4.1.** *Let $\widehat{p}_\theta$ be the likelihood function that arises from optimizing Equation 3 using the true Bellman likelihood operator. Then, $\widehat{p}_\theta$ satisfies*

$$Q^*(s,a) \geq \widehat{p}_\theta(s,a) \geq Q^*(s,a)\pi_\beta(a \mid s) ,$$

*for all $s \in \mathcal{D}$ and $a \in \mathcal{A}$ such that $Q^*(s,a) \geq \frac{1}{|\mathcal{A}|-1}$.*

We defer proof of the theorem to Appendix A. Note that our probabilities are conservative only over actions that have non-negligible Q-values. In practice, we do not see this as a problem as actions with negligible Q-values will not be chosen anyway. Overall, we show that while our objective looks very different from traditional value-based RL, our method still learns a conservative value function.

So far, we have shown that theoretically, our algorithm achieves similar theoretical properties as value-based RL methods, even though our objective looks closer to supervised learning. Next, we will compare our approach to other RL algorithms adapted from supervised learning such as filtered behavior cloning or return-conditioned supervised learning, and show why ours is beneficial.

**Filtered behavior cloning.** Filtered BC attempts to adapt supervised fine-tuning to non-expert datasets by only training on the top $\rho$-percent of trajectories by reward for $\rho \in [0,1]$. While natural, this harms sample efficiency as our method, like value-based RL methods, is also able to extract meaningful knowledge from low-reward trajectories.

**Return-conditioned supervised learning.** RCSL remedies the issues with filtered BC by conditioning on reward during training to extract multiple policies rather than just the expert one. However, we argue that RCSL still fails to learn from low-reward trajectories as well as our approach. Specifically, our approach and others that learn value functions can learn from suboptimal trajectories by stitching them into a better policy (Fu et al., 2020). However, Brandfonbrener et al. (2022) showed that RCSL cannot perform stitching in general, which limits its effectiveness at learning from suboptimal data.

## 4.3 Practical Implementation

Our objective in Equation 3 trains the model such that the predicted token probabilities match their Q-values. Our final step is to choose how to use these probabilities in the final learned policy. Prior work performs policy extraction that learns a policy regularized to be similar to the behavior policy (Peng et al., 2019; Kostrikov et al., 2021). Namely, it is well-known that a policy parameterization $\widehat{\pi}(a|s) \propto \pi_\beta(a|s) \exp(\beta Q^*(s,a))$ is a solution to the constrained optimization problem (Peng et al., 2019; Brandfonbrener et al., 2021)

$$\arg \max_\pi \ \mathbb{E}_{s \sim d^{\pi_\beta}, a \sim \pi} \left[ Q^*(s,a) \right] \quad \text{s.t.} \quad \mathbb{E}_{s \sim d^{\pi_\beta}} \left[ D_{\text{KL}}(\pi(\cdot \mid s) \,||\, \pi_\beta(\cdot \mid s)) \right] \leq \varepsilon ,$$

where $\beta > 0$ is a hyperparameter derived from the Lagrange multiplier. Recall that we have already approximated $\pi_\beta$ by optimizing an $\mathcal{L}_{CE}(\phi)$ over parameters $\phi$. Therefore, without any further training we can extract a policy whose probabilities of actions follow $\widehat{\pi}(a|s) \propto \pi_\phi(a|s) \exp(\beta p_\theta(a|s))$, which can be computed at inference time using our learned probabilities $p_\theta$, estimated behavior policy $\pi_\phi$, and tunable hyperparameter $\beta > 0$. Note that unlike prior works that explicitly require a policy

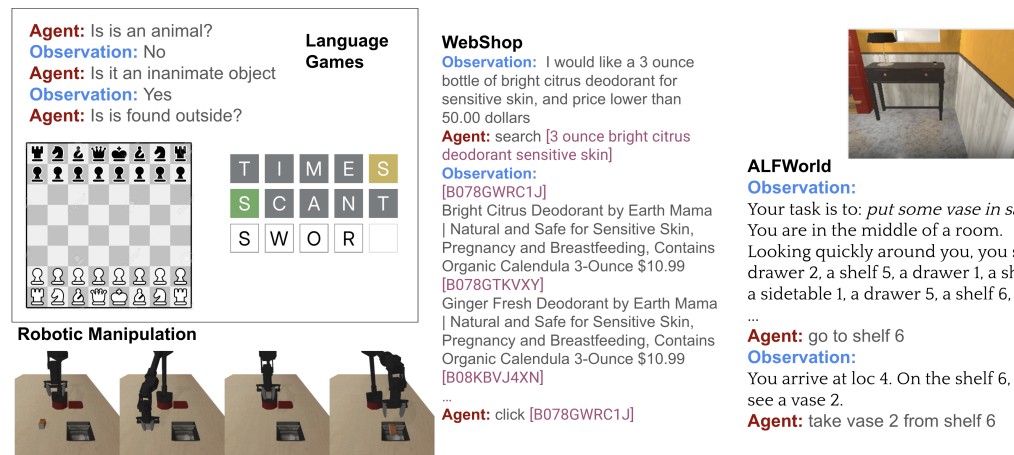

Figure 2: Overview of all the evaluated tasks, spanning both text and image inputs. Solving all the tasks effectively requires our algorithm to be able to be used to fine-tune LLMs, VLMs, and even robotics transformer models.

extraction step with additional training, our policy can be computed at inference-time. Namely, we can express our policy using only the probabilities that we had previously learned. An overview of our entire method is provided in Algorithm 1. We also provide implementation details such as hyperparameter selection for our experiments in Appendix B.

---

**Algorithm 1** Q-learning via Supervised Fine-tuning (Q-SFT)

---

**Require:** Dataset $\mathcal{D} = \{(s_i, a_i, r_i, s'_i)\}_{i \in [N]}$, hyperparameter $\beta > 0$
 1: Initialize $\phi, \theta, \bar{\theta}$ from pretrained model.
 2: *Optimize behavior policy:*
 3: **for** each gradient step **do**
 4:     Update $\phi \leftarrow \phi - \lambda_\phi \nabla_\phi \mathcal{L}_{\text{CE}}(\phi)$
 5: **end for**
 6: *Optimize likelihood model:*
 7: **for** each gradient step **do**
 8:     Update $\theta \leftarrow \theta - \lambda_\theta \nabla_\theta \mathcal{L}_{\text{WCE}}(\theta)$
 9:     Update target parameters: $\bar{\theta} \leftarrow (1 - \alpha)\bar{\theta} + \alpha\theta$
10: **end for**
11: *At inference time, policy probabilites become:* $\widehat{\pi}(a \mid s) \propto \pi_\phi(a \mid s) \exp(\beta \, p_\theta(a \mid s))$

---

## 5 Experiments

Our method combines aspects of both SFT and value-based RL training, and we therefore compare our method to state-of-the-art methods from both classes, evaluating:

(1) Whether our method improves over SFT methods by taking into account and optimizing over a multi-step task reward.

(2) Whether our method improves on the stability and performance of previously proposed value-based RL methods for training LLMs and VLMs.

(3) Whether our method is better able to benefit from the pretraining of large models than previously proposed multi-turn RL methods.

In this section, we perform a comprehensive empirical evaluation across a suite of different tasks to find positive answers to all the above questions.

### 5.1 Task Descriptions

Contrary to many existing applications of RL on language models, such as RLHF (Ouyang et al., 2022) or DPO (Rafailov et al., 2023), our proposed algorithm is tailored for offline RL on *multi-step*

tasks. Therefore, we consolidate a variety of existing benchmarks where a language model must make sequential decisions, arriving at the following suite of different tasks.

The first set of tasks include language games from the LMRL benchmark (Abdulhai et al., 2023), which is one of the first benchmarks tailored at evaluating offline RL for language generation.

**Chess.** This task uses a textual representation of the game of chess. The offline dataset consists of trajectories by Stockfish 15.1 simulating various player strengths as the agent, playing against another Stockfish engine with Elo 1200. The reward is 1 for a move that results in victory, 0 for a legal move and -1 for an illegal move. Our dataset consists of 625K trajectories of full games, in which the agent achieves an average return of 0.21.

**Wordle.** In the game of Wordle, the agent is given at most 6 attempts to guess a hidden 5-letter word. After each guess, the agent is told whether each letter in the guessed word is: (1) in the hidden word and in the right position (green), (2) in the hidden word but not in the right position (yellow), or (3) not in the hidden word (gray). The agent receives a reward of -1 after each incorrect guess. The dataset consists of 20K trajectories by a suboptimal heuristic policy that achieves an average return of -4.12, originally collected by Snell et al. (2022).

**Twenty Questions.** The final language task is the dialogue game of twenty questions, where the agent tries to guess what a hidden object is by asking a series of yes-or-no questions. The dataset consists of 100K conversations between an agent that is a guesser and the oracle that chooses the hidden word. The oracle chooses the hidden work uniformly at random from 158 unique objects. The guesser and the oracle are both simulated using `GPT3.5` (OpenAI, 2022), which is prompted to both generate questions and answer them factually. The agent receives a reward of -1 for each question that is not a correct guess, up to a minimum return of -20. The average return in the dataset is -17.3.

The next evaluation for fine-tuning LLMs as language agents — interactive web-based tasks that require using tools like search.

**WebShop.** an online shopping website environment where an agent processes unstructured text data (in the form of descriptions crawled from Amazon) to purchase a product given some initial user specifications. At the end, the agent receives a reward between 0 and 1 depending on the similarity between the purchased and ground-truth desired item. The benchmark consists of 12k initial user instructions, of which we randomly held out 100 for evaluation. With the remaining instructions, we generate a dataset of trajectories where we simulate an suboptimal agent by prompting `GPT3.5` with few-shot examples, following the prompts used by Yao et al. (2022).

Our method can be applied not only to language models, but also to multimodal models. In the next experiment, we study the performance of our method on vision-based navigation with VLMs.

**ALFWorld.** This is a popular text-based environment grounded in image observations (Shridhar et al., 2021). In this environment, the agent is tasked with solving one of 6 different task types, ranging from finding, moving, and manipulating different household objects within an embodied environment of 120 rooms. At each timestep, the agent observes a textual description of its location and surroundings with an analogous image, and chooses a text action from a set of admissible actions. In this environment, we sample 10k trajectories consisting of a random templated task description, and an attempted execution of the task within 30 timesteps by a prompted `GPT3.5` model for data collection. In the dataset, the agent only successfully accomplishes the task 34% of the time aggregated across all task types.

Finally, we also evaluate our method for training policies outside of language generation. Robotics is a popular domain in which offline RL has been proven effective for training per-token Q-values for continuous control (Singh et al., 2020; Chebotar et al., 2023). In these experiments, we do not leverage pretrained language models and simply test the effectiveness of the underlying RL algorithm.

**Robotic manipulation.** We consider the large-scale robotic manipulation control tasks from Singh et al. (2020). In the environment, the agent controls a 7-DoF mobile manipulator in front of a countertop surface to perform two types of tasks: pick up an object and place on the trap in front, or grab an object from the drawer. A sparse reward of 1 is received if the agent accomplishes the task. Following Singh et al. (2020), we collect 300k trajectories of randomized, scripted policies performing one of the two task types. The scripted policies achieve roughly 40% success rate.

We show illustrations of all the considered tasks in Figure 2.

| | language games | | | alfworld | | | | | |
|---|---|---|---|---|---|---|---|---|---|
| Method | Chess | Wordle | 20Q | Pick | Examine | Clean | Heat | Cool | Pick2 |
| ReAct | 0 | −4.96 | −13.2 | **45** | 19 | 17 | 7 | 12 | **24** |
| SFT | 0.11 | −3.81 | −17.3 | 38 | 15 | 0 | 11 | 0 | 18 |
| ILQL | 0.09 | −2.08 | −14.2 | 28 | 7 | 0 | 5 | 2 | 15 |
| Q-SFT (ours) | **0.15** | **−2.11** | **−13.1** | 39 | **21** | **19** | **14** | **18** | 21 |

Table 1: Average scores (for language games), and success rates (for ALFWorld tasks) across 100 independent evaluations. Our method performs best or near-best across the table, and competitively with prompting a much more complex model.

| Method | Score |
|---|---|
| ReAct | 0.60 |
| SFT | 0.55 |
| Offline ArCHer | 0.57 |
| Q-SFT | **0.63** |

| Method | Pick Object | Place Object Near Target |
|---|---|---|
| BC | 44 | 32 |
| CQL | 78 | 57 |
| QT | 92 | **68** |
| Q-SFT | **94** | 64 |

Table 2: Average score across 100 held-out instructions in WebShop. Our method performs best, even against prompting a much larger model.

Table 3: Success rate for 100 runs across robotic manipulation tasks. Our general method performs competitively with Q-transformer, a value-based RL method specifically designed for continuous control.

## 5.2 RESULTS

The goal of our empirical results is to show positive answers to all the proposed research questions. In order to do so, we evaluate multiple state-of-the-art supervised and value-based RL methods.

**Evaluation.** We compare our method Q-SFT against three classes of competing algorithms:

*Prompting*: ReAct (Yao et al., 2022) is an extension of chain-of-though prompting (Wei et al., 2023), where the pretrained language model is prompting to think and reason multiple steps in advance. We use `GPT3.5` (OpenAI, 2022) as the LLM, and `GPT4-V` (OpenAI, 2023) as the VLM.

*Supervised learning*: Supervised fine-tuning (SFT) on the offline dataset. In the case of using non-pretrained models, such as in the robotics task, we rename the method as behavior cloning (BC).

*Value-based RL*: Traditional offline RL algorithms that perform Q-learning to learn value functions. We consider several different algorithms, depending on the task at-hand. In the case of language games and ALFWorld, we evaluate ILQL (Snell et al., 2023), a popular approach for language generation. In WebShop, we consider an offline variant of ArCHer (Zhou et al., 2024) that performs best on the task from prior work. Finally, in robotics manipulation, we consider both CQL (Kumar et al., 2020) and Q-transformer (QT) (Chebotar et al., 2023), which are both popular and achieve state-of-the-art performance in continuous control.

Since the state-of-the-art LLMs and VLMs often only expose inference APIs, we instead train our considered methods on the `GPT2-medium` LLM, which consists of 345M parameters (Radford et al., 2019). For ALFWorld, which requires VLMs, we use `LLaVA-1.6` model as the pretrained model (Liu et al., 2023). Finally, for robotics, we use a randomly initialized Transformer architecture modeled after the popular `RT-1` model, which processes images and discretizes the action space into tokens (Brohan et al., 2023b). Note that for the Chess and Wordle tasks, because their state and action space are unlike natural language, we replace the pretrained weights with a random initialization. Therefore, these tasks, like robotics manipulation, only compares the methods in terms of their effectiveness as RL algorithms.

**Discussion** We report results of our evaluations in Tables 1, 2, and 3. For fine-tuning LLMs, Table 1 and 2 show that our Q-SFT method outperforms supervised learning, and different value-based RL methods, sometimes outperforming state-of-the-art by almost 30%. Similarly, for VLMs, as shown in Table 1, our approach beats supervised and value-based RL baselines, particularly on hard

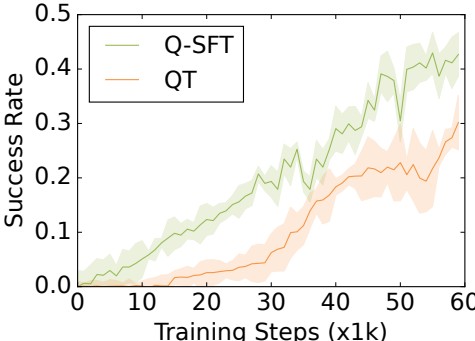
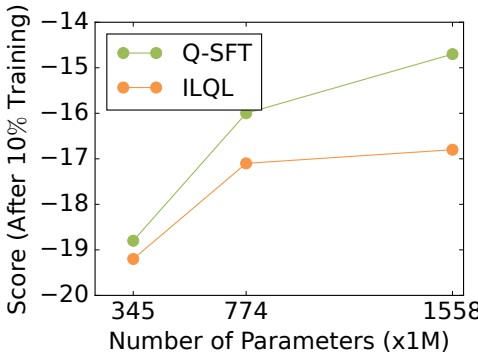

Figure 3: Success rate during initial training on the pick object task of the robotic manipulation benchmark. Though our method achieves similar final performance as Q-transformer, we perform much better on fewer samples.

Figure 4: Scores after training on $10\%$ of the offline dataset on the 20Q task, varying the size of the pretrained model. Our method benefits more from using more sophisticated pretrained models, suggesting our approach scales better.

task types with low success rate in the data. Our approach is also competitive with state-of-the-art prompting of `GPT4-V`, which contains about $30\times$ more parameters than the base models using during training. Finally, in Table 3, we see that even without leveraging any pretraining, our approach is competitive with state-of-the-art, suggesting that our objective is effective for learning values. This can be attributed to the fact that our underlying objective is more stable to optimize than traditional Q-learning ones, which require regression to non-stationary Bellman target values. Furthermore, in Figure 3, we show the learning curve for our approach compares favorably to QT, showing that our method learns more quickly and achieves better performance in the low-data regime.

Finally, we want to answer the last research question. We hypothesize that our approach benefits more from pretraining than existing value-based RL techniques, and verify this with an additional experiment. Specifically, we consider the 20Q task, and train both ILQL and Q-SFT policies against different models of increasing number of parameters, namely the `GPT2-large` and `GPT2-xl` models, which are $2\times$ and $5\times$ larger than `GPT2-medium` respectively. We also only train on $10\%$ of the original dataset, so that retaining prior knowledge from pretraining becomes crucially important. In Figure 4, we show the average return achieved by both methods across the different model sizes. We notice that for larger model sizes, Q-SFT significantly outperforms ILQL, implying a positive answer to the research question that our method retains knowledge acquired during pretraining better than existing value-based RL.

## 6 DISCUSSION

In this paper, we present Q-learning via Supervised Fine-Tuning (Q-SFT), a new offline RL algorithm where Q-values are learned as probabilities in an objective that looks like supervised fine-tuning. Because of this, our objective can be directly optimized over the logits of pretrained LLMs or VLMs To our knowledge, this is the first algorithm that can perform value-based RL fine-tuning without requiring any changes to the architecture, such as adding in new value heads. This has a number of important benefits. First, our objective is an instance of weighted cross-entropy, which has been shown by prior works to be more stable to train than traditional value-based RL methods that require regression towards non-stationary target values. More importantly, our algorithm fully leverages the advantage of foundation models such as LLMs or VLMs, as our algorithm starts from the pretrained probabilities, as opposed to randomly-initialized values. Theoretically, we show that our probabilities are conservative estimates of the true value function. Empirically, we compare our approach against strong supervised and value-based RL baselines on a variety of different tasks requiring LLMs, VLMs, and even robotics transformers. As future work, we aim to use our approach to also fine-tune vision-language-action (VLA) models, where we expect to see even greater benefit (Kim et al., 2024). Another more interesting direction for futher investigation, is whether our method can be adapted to also work online, as many recent works have considered online Rl optimization of language agents (Zhou et al., 2024).

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

# A  THEORETICAL PROOFS

Here, we provide a proof of Theorem 4.1. Recall that we are optimizing the following objective:

$$\mathcal{L}_{\text{WCE}}(p) = \mathbb{E}_{(s,a,r,s')\sim\mathcal{D}} \left[ \mathcal{B}^*\widehat{p}(a \mid s)\log p(a \mid s) + \sum_{a'\neq a} \frac{1 - \mathcal{B}^*p(a \mid s)}{|\mathcal{A}| - 1}\log p(a' \mid s) \right]. \quad (4)$$

Let us consider iteration $k$ of training. Setting the derivative of Equation 4 to 0, we obtain the following expression of $\widehat{p}^{k+1}$ in terms of $\widehat{p}^k$:

$$\forall s \in \mathcal{D}, a \in \mathcal{A}, \quad \widehat{p}^{k+1}(a \mid s) = \pi_\beta(a \mid s)\mathcal{B}^*\widehat{p}^k(a \mid s) + \sum_{a'\neq a}\pi_\beta(a' \mid s)\frac{1 - \mathcal{B}^*\widehat{p}^k(a' \mid s)}{|\mathcal{A}| - 1}. \quad (5)$$

**Lower-bound.** We will first show the lower-bound part of Theorem 4.1. Rearranging the above equation, we see that:

$$\frac{\widehat{p}^{k+1}(a \mid s)}{\pi_\beta(a \mid s)} = \mathcal{B}^*\widehat{p}^k(a \mid s) + \sum_{a'\neq a}\frac{\pi_\beta(a' \mid s)}{\pi_\beta(a \mid s)}\frac{1 - \mathcal{B}^*\widehat{p}^k(a' \mid s)}{|\mathcal{A}| - 1}. \quad (6)$$

Hence, we see that

$$\frac{\widehat{p}^{k+1}(a \mid s)}{\pi_\beta(a \mid s)} \geq \mathcal{B}^*\widehat{p}^k(a \mid s) = r(s,a) + \gamma\mathbb{E}_{s'\sim P(\cdot|s,a)}\left[\max_{a'}\frac{\widehat{p}^k(a' \mid s')}{\pi_\beta(a' \mid s')}\right],$$

where we substitute the definition of $\mathcal{B}^*$. Finally, taking the fixed point of the above expression yields,

$$\frac{\widehat{p}(a \mid s)}{\pi_\beta(a \mid s)} \geq Q^*(s,a) \Rightarrow \widehat{p}(a \mid s) \geq \pi_\beta(a \mid s)Q^*(s,a),$$

as desired.

**Upper-bound.** Now, we show the upper-bound part of Theorem 4.1. Assume that

$$\mathcal{B}^*\widehat{p}^k(a \mid s) \geq \frac{1}{|\mathcal{A}| - 1}.$$

Then, we can solve for the bound:

$$(1 - \pi_\beta(a \mid s))\,\mathcal{B}^*\widehat{p}^k(a \mid s) - \sum_{a'\neq a}\pi_\beta(a' \mid s)\frac{1 - \mathcal{B}^*\widehat{p}^k(a' \mid s)}{|\mathcal{A}| - 1}$$

$$= \sum_{a'\neq a}\pi_\beta(a' \mid s)\left(\mathcal{B}^*\widehat{p}^k(a \mid s) - \frac{1 - \mathcal{B}^*\widehat{p}^k(a' \mid s)}{|\mathcal{A}| - 1}\right)$$

$$\geq \sum_{a'\neq a}\pi_\beta(a' \mid s)\left(\frac{1}{|\mathcal{A}| - 1} - \frac{1 - \mathcal{B}^*\widehat{p}^k(a' \mid s)}{|\mathcal{A}| - 1}\right) \geq 0.$$

This means that we have,

$$\widehat{p}^{k+1}(a \mid s) = \pi_\beta(a \mid s)\mathcal{B}^*\widehat{p}^k(a \mid s) + \sum_{a'\neq a}\pi_\beta(a' \mid s)\frac{1 - \mathcal{B}^*\widehat{p}^k(a' \mid s)}{|\mathcal{A}| - 1}$$

$$= \mathcal{B}^*\widehat{p}^k(a \mid s) - (1 - \pi_\beta(a \mid s))\,\mathcal{B}^*\widehat{p}^k(a \mid s) + \sum_{a'\neq a}\pi_\beta(a' \mid s)\frac{1 - \mathcal{B}^*\widehat{p}^k(a' \mid s)}{|\mathcal{A}| - 1}$$

$$\leq \mathcal{B}^*\widehat{p}^k(a \mid s).$$

Hence, we see that

$$\frac{\widehat{p}^{k+1}(a \mid s)}{\pi_\beta(a \mid s)} \leq \frac{\mathcal{B}^*\widehat{p}^k(a \mid s)}{\pi_\beta(a \mid s)} = \frac{1}{\pi_\beta(a \mid s)}\left(r(s,a) + \gamma\mathbb{E}_{s'\sim P(\cdot|s,a)}\left[\max_{a'}\frac{\widehat{p}^k(a' \mid s')}{\pi_\beta(a' \mid s')}\right]\right).$$

Finally, taking the fixed point of the above expression yields the desired $\widehat{p}(a \mid s) \leq Q^*(s,a)$. This completes the proof.

# B    IMPLEMENTATION DETAILS

We use the hyperparameters reported in Table 4. All algorithms were trained on a single TPUv3 on Google Cloud until convergence.

| Hyperparameter | Chess | Wordle | 20Q | WebShop | ALFWorld |
|---|---|---|---|---|---|
| $\beta$ | 8.0 | 4.0 | 1.0 | 1.0 | 1.0 |
| $\gamma$ discount factor | 0.99 | 0.99 | 0.95 | 0.9 | 0.95 |
| Batch size | 128 | 128 | 128 | 128 | 128 |
| Target network update $\alpha$ | 0.005 | 0.005 | 0.005 | 0.01 | 0.01 |
| Number of updates per iteration | 60 | 60 | 60 | 50 | 50 |
| Number of iterations | 100 | 100 | 100 | 200 | 200 |
| $\lambda_\phi$ learning rate | 1e-4 | 1e-4 | 1e-4 | 2e-4 | 3e-4 |
| $\lambda_\theta$ learning rate | 1e-4 | 1e-4 | 1e-4 | 2e-4 | 1e-4 |

Table 4: Hyperparameters used during training Q-SFT in our experiments.

As shown in Table 4, most hyperparameters were held the same except for $\beta$, where larger $\beta$ results in a more deterministic policy. In practice, we only had to increase $\beta$ for tasks with restricted action spaces (such as games). Hence, we can conclude that in most practical tasks, our method does does not require much hyperparamter tuning to perform well.

