# OpenReview forum: "Q-SFT: Q-Learning for Language Models via Supervised Fine-Tuning"
_ICLR.cc/2025/Conference — ICLR 2025 Poster_

### Official Review · Reviewer_zr9o · 2024-11-02

**Soundness:** 3
**Presentation:** 3
**Contribution:** 3
**Rating:** 6
**Confidence:** 3

**Summary:**

This paper proposes Q-SFT, a new approach that enables fine-tuning of LLMs and VLMs within an offline RL framework but using a supervised learning-style loss function. The method matches or outperforms previous approaches across a range of LM fine-tuning benchmarks.

**Strengths:**

- The motivation is clear: the instability of the Q-learning objective is a well-known challenge in the RL literature.
- The proposed method is straightforward and demonstrates strong empirical performance, with improvements in sample efficiency and scalability over previous methods.
- The evaluation covers a wide range of benchmarks.

**Weaknesses:**

It seems there are some limitations to the proposed approach that are not addressed in the paper.
- During inference, the approach requires double the number of forward passes—one for the original pretrained LM and one for the fine-tuned LM. This can be challenging in practical scenarios where real-time decision-making is essential, such as robotic manipulation.
- In Theorem 4.1., the lower bound for the learned p_theta is Q* multiplied by pi_beta. If pi_beta is small, the resulting pi_theta can be overly conservative compared to the true Q*.

**Questions:**

.

---

> ### Author Response · Authors · 2024-11-22
> **Response to Reviewer zr9o**
>
> Thank you for our review. You raised several important points that we aim to address below.
>
> **Slow inference**
>
> You are correct in that we require running an inference step on two different models, which is double the number of forward passes as traditional supervised learning. However, such forward passes can be done in parallel, which would not add to the required computation time. We would also like to point out that existing approaches that also consider value-based RL training also require an equal number of forward passes [1].
>
> [1] https://arxiv.org/pdf/2206.11871
>
> **Overly conservative**
>
> This is also a correct observation based on our theoretical analysis. However, we would like to point out that we simply show a lower-bound that is not guaranteed to be tight. In practice, across a wide range of different benchmarks, we see that when $\pi_\beta$ is poor, our method still performs very well and is not shown to be overly conservative.

---

### Official Review · Reviewer_ChGQ · 2024-11-02

**Soundness:** 3
**Presentation:** 3
**Contribution:** 4
**Rating:** 8
**Confidence:** 4

**Summary:**

This paper proposes a new RL training method for LLMs that take advantage of the pretraining. It is based on Q-learning, but instead of attaching a new value head, the method uses the existing token probabilities as value function. There is some theoretical guarantees and diverse experimental results involving multi-steps interactions.

**Strengths:**

- Novelty: the method seems to be quite novel. The way it works is quite different from a typical Q-learning where L2 loss is used. Instead it relies on a cross-entropy loss that is modified. Given the importance of RLHF in LLMs and the need of multi-step interaction, a study like this can be impactful and important.
- The experiments are quite diverse, ranging from LLM-based text tasks to VLM-based robotic manipulation. It is also compared to a diverse set of baselines, which helps to solidify the claims.
- There is a proof about a theoretical guarantee, which is always nice to have. The method also leads to an improvement in performance in practice too, where the gap increasing as the model scales. Given the lack of value-based approaches in LLMs, it would be interesting to see if this method will adopted widely.
- Paper is well written and easy to follow.

**Weaknesses:**

- I felt like the writing of the experimental results were a bit too short. It would have been good to have more ablation studies to understand why the method works, and in which situations it is more optimal. For example, since the main motivation is that the value head is not initialized from scratch, what will happen if we do that but keep the loss the same. Also there is not much on the training details, such as the number of training steps etc. I think the other parts can be condensed to make space.
- Another important part that was too short was the method itself. The method is introduced between L230-261, which is about only half page. It doesn’t give enough explanation about why this method should work, so it was quite hard to understand it. For example, can you give more insight into why the proposed method should be more stable?
- Reuse of the pretrained weights is emphasized as the main motivation, but some of the tasks actually train the model from scratch and others are not in natural language, which is a bit conflicting. There is still an improvement when trained from scratch, which is good, but there is a lack of explanation why it works better.

**Questions:**

- Some typos: L39 citation, L84 sentence, L194 “performing”, Fig2 top left,
- Eq2 should have a negative sign? Also tab1 -2.08 should be bold, not -2.11.
- The figure 1 was bit too small and hard to read.
- L153 defines RL as MDP, but what about POMDP?
- About the theoretical guarantee, how good is the lower bound? A probability from a policy can be quite low for large vocabulary, in which case then the bound is not that tight?
- How many models needed to be loaded during training? There is the main model, also a target model and the original model. So is that 3? How does that affect memory usage?
- Given that some baselines use different base models, it would be nice to put them in the table. This will help if they perform better because their base model is stronger or not.
- In fig3, the error bars correspond to different seeds?

---

> ### Author Response · Authors · 2024-11-22
> **Author Response to Reviewer ChGQ**
>
> Thank you for your review! We completely understand your concerns regarding writing. To address them, we have added a discussion of training details to Appendix B of our updated paper, and added some more prose in motivating our method. Furthermore, we want to address the questions you posed in our review below:
>
> **Theoretical guarantee**
>
> The lower-bound is simply there to show that our method will not become arbitrarily conservative by severely underestimating the true value function. We agree that this lower-bound may not lead to good performance when the behavior policy is very poor. However, we do not notice this to be a problem in practice, as our method significantly outperforms the behavior policy across a wide range of tasks.
>
> **Memory usage**
>
> That is correct in that we require $3$ copies of the model. This is similar to competing value-based RL approaches such as ILQL [1]. We agree with the reviewer that  this is still a downside of using value-based RL to fine-tune LLMs, and are working on modifications of our algorithm to reduce memory usage.
>
>
> [1] https://arxiv.org/pdf/2206.11871

---

> > ### Comment · Reviewer_ChGQ · 2024-12-03
> >
> > Thanks for the response. I will maintain my score.

---

### Official Review · Reviewer_WNH5 · 2024-11-04

**Soundness:** 3
**Presentation:** 1
**Contribution:** 3
**Rating:** 6
**Confidence:** 4

**Summary:**

This paper proposes a method that leverages the LLM logits learned via supervised fine-tuning to approximate the Q-learning objective. Authors argue that traditional Q-learning setup suffers from discarding the logit head, and learning a new head for Q values, so the proposed method directly translates the learned logits to Q values, with theoretical analysis on its bound and approximation. Experiments include text-based games, image-based navigation and robotic manipulation, and show some benefits of the proposed method over the prior value-based RL method.

**Strengths:**

- Using probability prediction in the Q-learning objective is cool, and it sounds particularly beneficial for large vocabulary/action space.
- The implementation of the method is straightforward: learning two models using different objectives, and use both at inference time

**Weaknesses:**

- I found the pitch very confusing: it emphasizes multi-turn problems like dialogue, but the actual math formulation treats each turn as a single token in an utterance (i.e., the action space is the vocabulary space). The generation process is to generate an utterance token by token, which is conventional, and it’s confusing where the multi-turn setup plays a role other than using this generative model repeatedly to generate different utterances at different parts of the dialogue. In other words, the value function here is not the reward per utterance as stated in the intro (line 074), and question answering is also not a single-step problem as the answer is also generated token by token.
- Experimental setup seems simplified in terms of 1) for language-based tasks, the game nature makes the effective vocabulary very small and unambiguous, 2) the well-defined reward function setup in text games do not translate into practical problems, 3) why no supervised learning results from prior work as a direct comparison?; 4) most of the datasets already have RL method outperform supervised learning which is contradictory to the intro “offline RL fine-tuning of LLMs does not reliably outperform supervised fine-tuning”. Therefore, it’s unclear whether the improvement brought by the proposed method should be contrasted to what exactly.
- The writing can be further improved in terms of clarity, accuracy, and fixing typos. A few examples below:
   - Table 1: number from the proposed method is bold in the Wordle column, but it’s not the best performance method
   - Define h used in line 162
   - Prime should be on a, instead of argmax in line 218
   - Consistently use “RL” instead of having “RI” occurred
   - Citation should be within a pair of parentheses: line 037, 039

**Questions:**

Please see the previous section.

Response to the rebuttal: (posted here since the rebuttal ended)

Thanks for all the clarification and discussion -- I raised my scores accordingly. It would be very valuable to revise the paper to address the points we discussed, especially the clear definition on step vs turn, the potential to scale better than existing RL approaches, the setup and comparable work of SFT baselines, motivation to alleviate the downsides of using RL, etc.. The rebuttal was helpful to better understand this work, and it would definitely be better if readers only need to read the paper itself to sufficiently appreciate this work.

---

> ### Author Response · Authors · 2024-11-22
> **Author Response to Reviewer WNH5**
>
> Thank you for your review. We aim to address the concerns  you had with our work. Please let us know if there is anything else we can clarify!
>
> **Multi-turn dialogue**
>
> The problems we study are “multi-turn dialogues” because our agent must respond to utterances by other agents, where each “turn” consists of a concatenation of all utterances made thus far (including by participants that are not the agent). It is true that each utterance in a turn is not an isolated action, but rather consists of a list of tokens/words that are each actions. Hence, each utterance consists of multiple steps of generation, equal to the number of tokens in the utterance. This is standard in formulating language generation as MDPs, where actions are considered on a token-level, and concatenated to form utterances [1].
>
> However, we believe there is still a big distinction between generating a single utterance as a response to a question, and what we study, which involves generating multiple utterances, each responding to what other agents have said thus far. Namely, in the former, the outcome/reward of the utterance is immediately observed, whereas in the latter, the agent may need to output multiple utterances before receiving an outcome/reward. This is the key difference and why we believe the latter requires RL optimization of some long-term objective.
>
> We understand where confusion may arise, as we call these “single-turn” problems “single-step” even though they still require multiple steps of token-level generation. In the updated paper, we make a distinction between a “turn” and a “step,” where “turn” involves producing a single utterance, and “step” a single token in an utterance. Therefore, “single-turn dialogues,” such as answering a question, still involve multiple steps of generation. And in “multi-turn dialogues”, the agent engages in a cycle of outputting an utterance, observing what others say, then responding by outputting a new utterance. We hope this alleviates the confusion you had with our work.
>
> [1] https://arxiv.org/pdf/2210.01241
>
> **Experimental setup seems simplified**
>
> We consider a diverse range of publicly available benchmarks to evaluate our approach. While each individual task may have certain limitations, we believe that considering them in aggregate shows that our approach achieves strong empirical results. We will also address each point that the reviewer raised sequentially:
>
> 1. It is true that some language games such as Wordle and Chess involve a limited vocabulary of possible actions, but we also include 20Q which allows for open-ended language generation.
>
> 2. Additionally, while the games have a well-defined reward, we also evaluate our method on tasks such as navigation or tool use, where the reward directly translates to “success” or “failure.” We believe this is a very practical formulation of reward that can be applied generally.
>
> 3. Our baseline SFT/BC exactly corresponds to supervised learning, and achieves performance that is similar to that reported in prior works.
>
> 4. It is true that RL often still does outperform BC, but usually very marginally. Compared to existing domains such as robotics, the gap between RL and BC on suboptimal demonstrations seems surprisingly small. We believe this is important to address, and is the main motivation behind our work. We have updated the language in our paper to make this more clear.
>
> Finally, we believe we have considered all the relevant and publicly available benchmarks to evaluate our method. If you are aware of any others, we would be happy to add them in our updated paper!

---

> ### Comment · Reviewer_WNH5 · 2024-11-27
>
> Thanks to authors for the rebuttal. I still have confusions regarding the reply, since I don't feel my questions are being directly answered:
>
> Regarding my question 1 *for language-based tasks, the game nature makes the effective vocabulary very small and unambiguous*: The author response is "we also include 20Q which allows for open-ended language generation". I want to emphasize that my point is the small size of the *effective* vocabulary, which is determined by the language game setup, not by the form of the task. For example, 20Q dataset itself is to guess a hidden objective with several yes-no questions, but the dataset annotation itself is composed of very short yes-no questions (the viewer in this link gives examples: https://huggingface.co/datasets/clips/20Q). If we want to be super rigorous, we could compare the vocab size of all model generations on 20Q, and I would be surprised if this vocab size matches with the standard vocab size of truly open-ended generation in a non-game setup.
>
> Regarding my question 2 *the well-defined reward function setup in text games do not translate into practical problems*: The author response is "we also evaluate our method on tasks such as navigation or tool use, where the reward directly translates to “success” or “failure.” We believe this is a very practical formulation of reward that can be applied generally." I was trying to think from the perspective that practical problems may not have well-defined reward functions that would give an informative/correct reward every time. This is different from checking the formulation of the reward function, but more about wether the method would work with not-so-well-defined reward function. Take navigation as an example, navigation problems in ALFRED datasets study agents in a simulated environment, and the success/failure rate makes sense. By practical problems, I am thinking about no given pre-built grid-map of the scene and thus noisy rewards to work with. More clearly, I'd like to learn if the proposed method would work well in such setup. (To clarify, I'm fine with methods that would only work well with well-defined reward functions -- I just advocate for being clear and straightforward if this is the case.)
>
> Regarding my question 3 *why no supervised learning results from prior work as a direct comparison?*: the reply is "Our baseline SFT/BC exactly corresponds to supervised learning, and achieves performance that is similar to that reported in prior works". It's clear that SFT/BC exactly corresponds to supervised learning in your paper, but my point is why not directly citing prior work? I suggest to directly cite the number from prior work instead of reporting similar numbers from own experiments, so that readers can directly find the comparable prior work. Please correct me if I miss anything -- I am super curious about why SFT is so bad on alfword dataset (0 success on "Clean" and "Cool" tasks), so I'd like to check SFT prior work on this. The SFT baseline in alfword dataset paper seems to have not-so-similar results: 12 and 21 on "Clean" and "Cool" tasks (seq2seq baseline in table 2 https://arxiv.org/pdf/2010.03768), so I wonder if I miss something fundamental.
>
> Regarding my question 4 *most of the datasets already have RL method outperform supervised learning which is contradictory to the intro “offline RL fine-tuning of LLMs does not reliably outperform supervised fine-tuning”. Therefore, it’s unclear whether the improvement brought by the proposed method should be contrasted to what exactly.*: To be extremely clear on why I ask this question, I want to understand the main argument of this paper: the proposed method is better than prior RL method, or the proposed method is better than prior SFT/BC method. The author response is about the gap between RL and BC ("It is true that RL often still does outperform BC, but usually very marginally. Compared to existing domains such as robotics, the gap between RL and BC on suboptimal demonstrations seems surprisingly small. We believe this is important to address, and is the main motivation behind our work."), which I don't doubt as the result table in this work shows so. I wonder why not directly argue that the proposed method performs better than prior RL method. Given that prior RL method outperforms SFT already, just arguing the proposed method is better than SFT seems strange, as it may imply that the proposed method is behind prior RL method.
>
> I'd like to further clarify that I have nothing against training and evaluating on publicly available benchmarks, which may be constrained in different ways -- I'm trying to better understand some general claims in this work by asking questions about 1) what exactly is the improvement compared to prior work; 2) wether we can expect the same improvement if we were to apply the proposed method to tasks without those constrains.

---

> ### Author Response · Authors · 2024-12-02
> **Author Response to Reviewer WNH5**
>
> Thank you for raising these additional concerns, as they are very valuable feedback! Let us know if you have any additional questions or concerns that we can address.
>
> **Effective vocabulary is small**
>
> We agree that the language tasks we evaluate on do not cover the diversity of truly open-ended language generation. We believe this is more a limitation of the available benchmarks than of our method. Our method is meant to alleviate the downsides of traditional value-based RL fine-tuning of LLMs and VLMs, namely in operating directly on the pre-trained model rather than requiring modifications to architecture. This means our approach should, in theory, scale better than existing RL approaches by leveraging more sophisticated, pre-trained models more effectively. However, we agree with the reviewer that it is important to still evaluate our method on open-ended language benchmarks as they become available.
>
> **Well-defined reward**
>
> We agree with the reviewer that some rewards can be much harder to define than the ones we consider in our experiments. Our approach is still an RL method, and meant to be used on problems that can be formulated as MDPs. By nature, this means some reward function must exist, even if it is a noisy one. We will make this more clear when introducing our method.
>
> However, we would like to point out that many advancements with LLM-as-a-judge literature as shown that when a reward function doesn't yet exist for the task (such as open-ended dialogue), an LLM can act as a reasonable reward function [1]. This gives us cause to believe that requiring a reward function will not become a serious limitation.
>
> 1. https://arxiv.org/pdf/2306.05685
>
> **Supervised learning results**
>
> Our SFT results are considered in a different setting than the ones you pointed out. Namely, we do not assume access to expert demonstrations, but rather generate trajectories from a GPT-4 policy (that does not always succeed). This is simply because in the setting of having access to expert demonstrations, prior work has shown that there is no benefit to using RL approaches [1]. Therefore, when we evaluate SFT, we consider our setting where the dataset contains suboptimal demonstrations to directly compare to RL methods. Our SFT results on ALFWorld are similar to prior works that also consider suboptimal data, so we have no reason to believe it is due to an implementation issue [2]. We understand how the confusion arose and will make this distinction clear.
>
> 1. https://arxiv.org/pdf/2204.05618
>
> 2. https://arxiv.org/pdf/2405.10292
>
> **Offline RL vs supervised fine-tuning**
>
> We understand and agree with your point. Our story should not focus on reliably outperforming SFT. Our motivation was that current limitations of RL in language tasks make most practitioners heavily prefer SFT over the likely marginal gains RL provides. Hence, our method is meant to alleviate the downsides of using RL (by using a novel, easier-to-train objective) while still outperforming existing approaches, so that practitioners will now have strong reason to prefer our approach over SFT. We will revise our work to focus on this point instead.

---

### Official Review · Reviewer_oykW · 2024-11-04

**Soundness:** 4
**Presentation:** 3
**Contribution:** 3
**Rating:** 8
**Confidence:** 4

**Summary:**

This paper proposed Q-SFT, a method that integrates Q-learning with SFT for LLMs in the multi-turn RL setting. To exploit both the representation and the logits in LLMs learned in the pretraining stage and bypass the poor scaling effect of TD-style objectives, the authors proposed a novel weighted cross-entropy loss that embeds the learning of the Q-value into the weights. In this way, the logits prior in the pretrained LLMs can be leveraged, and no new head is needed to learn the Q-values. Theoretical analysis has shown the guarantee of the new learning objective leading to a conservative estimation of the value function. Experiments on several scenarios have demonstrated the effectiveness and efficiency of the proposed method.

**Strengths:**

- The work targets at the multi-turn RL scenario for LLMs, which is vital for the development of complex reasoning and agentic use in LLMs.
- The proposed algorithm is backuped with a solid theoretical motivation.
- The experimenting scenarios have a broad coverage, and the performance is well-demonstrated.

**Weaknesses:**

Please see the Questions below.

**Questions:**

- While the theoretical analysis shown in Section 4.3 demonstrates that the Q value learned with Equation (3) has the bounds independent from the hyperparameter \gamma, I wonder what the ablating effect \gamma brings in practice: Will certain configuration of \gamma contribute the learning of a better Q-value estimation? This question has another motivation: In Equation (3), some of the probability mass is allocated to the dummy actions a', and the weight ratio between the actions in the demonstration trajectory and the dummy ones has one degree of freedom determined by \gamma. It would therefore be great to sweep \gamma to study the ablation effect of the weight ratio.
   - And by the way, how is the \gamma set in the demonstrated experiments?

- According to Equation (3), learning the Q-value function requires a learned behavior policy \pi_\beta in prior. How will the quality of the behavior policy impact the learning of the Q-values, and the performance of the ultimate policy? While I appreciate the experiments in Figure 4 where only 10% of the training data are used to get the \pi_\beta, I wonder how the performance grows with the quality of the initial behavior policy. For example, if the initial behavior policy is near perfect, will it be that the ultimate policy, although weighted with the learned Q-values, performs similarly with the initial one?

---

> ### Author Response · Authors · 2024-11-22
> **Author Response to Reviewer oykW**
>
> Thank you for your review. You raised several great points that we aim to respond to below.
>
> **Hyperparameter $\gamma$**
>
> The $\gamma$ hyperparameter is exactly the discount rate in traiditional RL problems. Therefore, it is set to $1$ or very close to $1$ across all of our problems. We added a detailed table of hyperparameters to Appendix B of our updated paper. It is true that $\gamma$ can be changed, but the result would be biased Q-values. Specifically, decreasing $\gamma$ can result in a more myopic policy that values immediate reward over future rewards. In practice, setting $\gamma$ very close to $1$ performs best. We made this more clear in our updated paper.
>
>
> **Quality of behavior policy**
>
> That is also an excellent point. We use the probabilities output by the initial behavior policy essentially as a starting point for our algorithm. Therefore, the better the initial policy, the less training is required for our algorithm to converge. One way that we demonstrate this is in Figure 4 by varying the base model, as more sophisticated models should behave as better initial behavior policies. We see that our approach converges much more quickly than standard RL approaches because we leverage the better performance of the initial behavior policy.

---

> > ### Comment · Reviewer_oykW · 2024-11-27
> >
> > Thank you for the rebuttal! The added configuration table in the appendix is especially helpful.
> > I particularly like the simplicity of the proposed method, and I believe it should be encouraged. I'm raising my score.

---

### Public Comment · ~Tadashi_Kozuno1 · 2024-11-27
**Questions about the proof of Theorem 4.1**

Hi. Thank you very much for the interesting paper. While reading the proof of Theorem 4.1, I was not able to follow some reasonings. It is very much appreciated if the authors could clarify.

1. How did you derive the inequality just below Eq (6) from Eq (6)? I guess $\mathfrak{B}^\star \widehat{p}^k$ is assumed to be less than 1, but it is not clear for me why it is less than 1.
2. What is exactly meant by "taking the fixed point" in the proof? For example, in Line 861, simply expanding $\widehat{p}^k / \pi_\beta$ according to the inequality in Line 861 does not seem to lead to the conclusion in Line 863.

Regarding the first question, I think $\mathfrak{B}^\star \widehat{p}^k \leq 1$ does not hold in general for any $k$. I can construct a counterexample with a two-state two-action MDP and uniform $\widehat{p}^1$. (I might be wrong though.)

Regarding the second question, suppose a simple case where both rewards and behavior policy are constant (say $r$ and $\pi$ everywhere). Then simply expanding $\widehat{p}^k / \pi_\beta$ in RHS of Line 861 according to the inequality in Line 861, I end up with something like $\sum_{n=1}^{k} \gamma^k \pi^{-k} r$, which diverges if $\gamma > \pi$. Am I missing something?

---

### Meta-Review · Area_Chair_FhdA · 2024-12-21

**Metareview:**

This paper proposes a new way to do RL in multi-turn dialogue setting where the use of RL is less common compared to the vast success of RL in single-turn case. The proposed approach frames Q-learning as a cross-entropy problem with suitably selected weights such that the trained policy can be used to recover conservative bounds on the Q-value.

While I think the mathematical reasoning here is not important, the reviewers appreciated the mathematical analysis. However, upon going over the mathematical derivations in the paper I noticed that certain steps were not justified. While equation 5 is correct, the next equation isn't justified. It along with the WCE loss assumes that $\mathcal{B}^\star p_\theta$ is in $[0, 1]$ but it isn't clear why this is the case even with assumption 4.1. In particular, if for some $a'$ we have $\pi_\beta(a' \mid s') = 0$, then for a uniform $p_\theta$ we have $\mathcal{B}^\star p_\theta = \infty$. Note that the public comment also points this out. Now, if we assume $\mathcal{B}^\star p_\theta \in [0, 1]$, then we get can complete the proof. These steps are missing in the proof but are as follows:

$\alpha^{(k+1)}(s, a) \ge r(s, a) + \gamma E_{s' \sim P(. | s, a)}[ \max_{a'} \alpha^{(k)}(s', a')]$,

where $\alpha^{(t)}(s, a) = \frac{p^{(t)}(a \mid s)}{\pi_\beta(a \mid s)}$ for any $t \in \mathbb{N}$. Now if you assume convergence that $\lim_{t\rightarrow \infty} \alpha^{(t)}(s, a) = \alpha(s, a)$, we get:

$\alpha(s, a) \ge r(s, a) + \gamma E_{s' \sim P(. | s, a)}[ \max_{a'} \alpha(s', a')]$,

using Bellman optimality,
$Q^\star(s, a) = r(s, a) + \gamma E_{s' \sim P(. | s, a)}[ \max_{a'} Q^\star(s', a')]$

and defining $\delta(s, a) = \alpha(s, a) - Q^\star(s, a)$ and $\delta = \inf_{s, a}\delta(s, a)$, one gets:

$\delta(s, a) \ge \gamma E_{s' \sim P(. | s, a)}[ \max_{a'} \alpha(s', a') - \max_{a'} Q^\star(s', a')] \ge \gamma E_{s' \sim P(. | s, a)}[ \min_{a'} \left( \alpha(s', a') - Q^\star(s', a')\right)] \ge \gamma \delta$,

where the last line uses $\max_x f(x) - \max_x g(x) \ge \min_x (f(x) - g(x))$. Finally, taking inf on LHS means that $\delta \ge \gamma \delta$, which implies $\delta \ge 0$ if $\gamma \in (0, 1)$. Or, $\alpha(s, a) \ge Q^\star(s, a)$. I'd request authors to add justification for why $B^\star p \in (0, 1)$ and these missing steps.

Strengths:
1. Method is simple
2. Experiments are presented for several domains
3. Multi-turn LLM reasoning is an important topic and the use of RL for this has been somewhat less successful. This approach proposes a solution for this.
4. Reviewers appreciated the theoretical result but see below.

Weakness:
1. The theoretical results are at best unjustified and at worst potentially incorrect
2. Requires 3 copies of the model. This makes it expensive in practice as even 2x increase in memory is costly and makes a given approach undesirable. This is important especially given the motivation of the authors to make the Q-learning approach user-friendly.
3. The reviewer ChGQ requested results where the value head is randomly initialized but the authors did not include this study in response. I understand that experiments can be hard to complete during the rebuttal period but this one would have been nice to see.

Overall, my biggest concern is the mathematical foundation of the approach. However, given that the approach works well empirically and is simple and should easy to try, I am leaning towards a weak accept.

**Additional Comments On Reviewer Discussion:**

Reviewers raised the following concerns:

1. Confusion about what turn means (reviewer WNH5). The authors properly addressed this.

2. Effective vocabulary is small (reviewer WNH5). The authors agreed that their experiments do not cover such challenges but that this was more of an artifact of existing benchmarks.

3. Practical effectiveness (3 copies of the model and 2 inference calls): The authors agreed that this is a concern.

4. There were additional writing suggestions that the authors addressed reasonably.

My main concerns are (3) and the theoretical justification the latter which reviewers did not raise. I am still leaning towards acceptance cause there are algorithms such as PPO which are easy to show that they fail theoretically with simple counter-examples but are nevertheless effective in practice. However, the paper shouldn't have steps that are missing justification or at worse wrong maths. So, I request authors to either add justifications or assumptions, or remove the maths as it isn't necessary.

---

### Decision · Program_Chairs · 2025-01-22

Accept (Poster)